# CRL4$^{Cdt2}$ ubiquitin ligase regulates Dna2 and Rad16 (XPF) nucleases by targeting Pxd1 for degradation

**Jia-Min Zhang**[1¤], **Jin-Xin Zheng**[1], **Yue-He Ding**[1], **Xiao-Ran Zhang**[1], **Fang Suo**[1], **Jing-Yi Ren**[1], **Meng-Qiu Dong**[1,2], **Li-Lin Du**[1,2]*

1 National Institute of Biological Sciences, Beijing, China, 2 Tsinghua Institute of Multidisciplinary Biomedical Research, Tsinghua University, Beijing, China

¤ Current address: Massachusetts General Hospital Cancer Center, Harvard Medical School, Boston, Massachusetts, United States of America
* dulilin@nibs.ac.cn

**Data Availability Statement:** All relevant data are within the manuscript and its Supporting Information files.

## Abstract

Structure-specific endonucleases (SSEs) play key roles in DNA replication, recombination, and repair. SSEs must be tightly regulated to ensure genome stability but their regulatory mechanisms remain incompletely understood. Here, we show that in the fission yeast *Schizosaccharomyces pombe*, the activities of two SSEs, Dna2 and Rad16 (ortholog of human XPF), are temporally controlled during the cell cycle by the CRL4$^{Cdt2}$ ubiquitin ligase. CRL4$^{Cdt2}$ targets Pxd1, an inhibitor of Dna2 and an activator of Rad16, for degradation in S phase. The ubiquitination and degradation of Pxd1 is dependent on CRL4$^{Cdt2}$, PCNA, and a PCNA-binding degron motif on Pxd1. CRL4$^{Cdt2}$-mediated Pxd1 degradation prevents Pxd1 from interfering with the normal S-phase functions of Dna2. Moreover, Pxd1 degradation leads to a reduction of Rad16 nuclease activity in S phase, and restrains Rad16-mediated single-strand annealing, a hazardous pathway of repairing double-strand breaks. These results demonstrate a new role of the CRL4$^{Cdt2}$ ubiquitin ligase in genome stability mainte-nance and shed new light on how SSE activities are regulated during the cell cycle.

## Author summary

Structure-specific endonucleases are enzymes that process DNA intermediates generated in DNA replication, recombination, and repair. Proper regulation of these enzymes is crit-ical for maintaining genome stability. Dna2 and XPF are two such enzymes present across eukaryotes, from yeasts to humans. Here, we show that in the fission yeast *Schizosaccharo-myces pombe*, the activities of Dna2 and Rad16 (the equivalent of human XPF) are tempo-rally controlled during the cell cycle by the CRL4$^{Cdt2}$ ubiquitin E3 ligase. In the S phase of the cell cycle, CRL4$^{Cdt2}$ promotes the degradation of Pxd1, which is an inhibitor of Dna2 and an activator of Rad16. Through targeting Pxd1 for degradation, CRL4$^{Cdt2}$ increases the activity of Dna2 in S phase and is important for the normal S-phase function of Dna2. Meanwhile, the degradation of Pxd1 reduces the activity of Rad16 in S phase, and curtails Rad16-dependent single-strand annealing, a mutagenic DNA repair pathway. Our

**Funding:** This work was supported by grants to L.-L.D and M.-Q.D from the Ministry of Science and Technology of China and the Beijing Municipal Government. The funders had no role in study design, data collection and analysis, decision to publish, or preparation of the manuscript.

**Competing interests:** The authors have declared that no competing interests exist.

findings uncover a new mechanism regulating two important endonucleases during the cell cycle, and reveal a new way of coordinating endonucleases to safeguard genome stability.

## Introduction

Structure-specific endonucleases (SSEs) play crucial roles in the maintenance of genome stability by processing DNA intermediates during DNA replication, recombination, and repair [1,2]. Tight control of these nucleases is critical for accurately processing specific DNA structures without causing unnecessary DNA lesions [3,4]; however, the molecular mechanisms underlying the regulation of these nucleases have not been sufficiently revealed.

Scaffold proteins that bind to and regulate multiple SSEs have emerged as pivotal regulators of SSEs. For example, in human cells, the scaffold protein SLX4 interacts with and activates three SSEs: XPF-ERCC1, MUS81-EME1, and SLX1 [5–10]. In the fission yeast *Schizosaccharomyces pombe*, we previously identified a scaffold protein Pxd1 that binds to two SSEs, Rad16-Swi10 (equivalent of human XPF-ERCC1) and Dna2-Cdc24, and showed that Pxd1 activates the nuclease activity of the former but inhibits the nuclease activity of the latter [11]. For simplicity, hereafter we will refer to these two *S. pombe* SSEs by the names of their catalytic subunits, Rad16 and Dna2, respectively.

Both Rad16 and Dna2 are involved in multiple DNA metabolism processes. Rad16 acts together with Pxd1 in single-strand annealing (SSA) repair of double-strand breaks (DSBs), mating type switch, and the removal of Top1–DNA adducts [11–14], while Dna2 functions in DNA end resection, Okazaki fragment maturation, and the processing of stalled replication forks [15–22]. To properly fulfill their diverse roles, Rad16 and Dna2 are likely subject to regulation. In particular, regulation may be needed for the following two reasons. Firstly, cellular demands for SSE activities vary during the cell cycle. For example, during an unperturbed cell cycle, the need for the nuclease activity of Dna2 intensifies in S phase due to its role in Okazaki fragment maturation. Secondly, certain pathways that these SSEs are involved in can pose danger to the genome. For example, SSA can cause repeat-mediated genomic deletion and rearrangement [23–25]. In principle, Pxd1 can serve as a regulatory hub to allow fine-tuning of the SSE activities of Dna2 and Rad16, so that varying demands can be better met and threats caused by SSE-associated processes can be mitigated. However, it is unclear whether and how Pxd1 is regulated.

CRL4$^{Cdt2}$, a PCNA-dependent E3 ubiquitin ligase composed of Rbx1/Roc1, cullin 4, Ddb1, and Cdt2, prevents DNA rereplication by targeting the replication licensing factor Cdt1 for degradation in multiple species including *S. pombe* and humans [26–34]. In addition, it also ensures a sufficient supply of dNTP in S phase in *S. pombe* by promoting the degradation of Spd1, an inhibitor of the ribonucleotide reductase [35–38]. It was initially shown in vertebrates that protein ubiquitination and turnover mediated by CRL4$^{Cdt2}$ is dependent on PCNA and a special PCNA-binding motif termed the "PIP degron" in the substrates [39], and that the interaction between PCNA and the PIP degron is necessary for substrate recognition by CRL4$^{Cdt2}$ [40]. In *S. pombe*, CRL4$^{Cdt2}$-mediated degradation of Cdt1 and Spd1 also requires PCNA and PIP degrons [41,42], indicating that the PCNA-dependent mechanism is conserved from fission yeast to humans. With the identification of additional CRL4$^{Cdt2}$ substrates including P21, E2F, DNA polymerase η, Set8, Xic1, Epe1, P12, FBH1, Cdc6, thymine DNA glycosylase (TDG), and XPG [43–61], CRL4$^{Cdt2}$ has become recognized as a major regulator that controls the turnover of many proteins related to DNA replication and genome stability maintenance.

In this study, we show that the activities of Dna2 and Rad16 in *S. pombe* are regulated by CRL4$^{Cdt2}$ E3 ligase through degrading Pxd1 in S phase. The ubiquitination and degradation of Pxd1 in S phase is mediated by CRL4$^{Cdt2}$, PCNA, and a PIP degron on Pxd1, suggesting that Pxd1 is a substrate of CRL4$^{Cdt2}$. Preventing the degradation of Pxd1 leads to interference in the S-phase functions of Dna2 and unchecked activity of Rad16 in S phase. These results demonstrate that temporally controlled degradation of Pxd1 by CRL4$^{Cdt2}$ is a physiologically important mechanism to regulate the SSE activities of Dna2 and Rad16.

## Results

### The protein level of Pxd1 is reduced in S phase

Previously, we found that Pxd1 is a scaffold protein in the PXD (*pombe* XPF and Dna2) complex, and can activate the SSE activity of Rad16 but inhibit the SSE activity of Dna2 [11]. To explore whether Pxd1 is subject to regulation, we used live cell imaging to examine the fluorescence signal of YFP-tagged Pxd1 expressed from the endogenous promoter. Pxd1-YFP exhibited a nuclear localization pattern (Fig 1A), as expected from the roles of Pxd1 in regulating DNA nucleases in the nucleus. Interestingly, the fluorescence signal of Pxd1-YFP was readily observed in some but not all cells (Fig 1A). Particularly, Pxd1-YFP signal was much more infrequently seen in cells with septa than in cells without septa (Fig 1A and 1B). In *S. pombe*, S phase coincides with the presence of the septum [62,63]. Thus, the level of Pxd1 appeared to be reduced in the S phase.

To verify that Pxd1 is temporally regulated during the cell cycle, temperature-sensitive *cdc25-22* mutant was used to synchronize the cell cycle. When the cells were arrested in G2 phase by incubating at the restrictive temperature, strong fluorescence signal of Pxd1-YFP was present in virtually all cells (Fig 1C). Releasing these cells back into the cell cycle led to a notable reduction of the signal of Pxd1-YFP after 50 min, and a nearly complete loss of the signal at the 75 min time point, when the percentage of septum-containing cells peaked (Fig 1C). Immunoblotting analysis using endogenously TAP-tagged Pxd1 confirmed that the protein level of Pxd1 fluctuated during the cell cycle, being the lowest in S phase (Fig 1D).

Consistent with the above observations, hydroxyurea (HU) treatment also led to a strong reduction of the percentage of cells with visible fluorescence signal of Pxd1-YFP, presumably due to HU-induced S-phase arrest (Fig 1E). Cells expressing Pxd1-YFP from an exogenous *P81nmt1* promoter also exhibited a similar reduction of Pxd1 protein level upon HU treatment (S1 Fig), indicating that the S-phase reduction of Pxd1 does not require transcriptional regulation through its endogenous promoter.

### CRL4$^{Cdt2}$ mediates the ubiquitination and degradation of Pxd1 in S phase

Because CRL4$^{Cdt2}$ E3 ligase-mediated degradation is a known mechanism driving S-phase-specific protein downregulation in *S. pombe* [30,32,37,64,65], we examined whether this mechanism is behind the S-phase reduction of Pxd1. In *ddb1Δ* and *cdt2Δ* mutants, which respectively lack one of the subunits of CRL4$^{Cdt2}$, the signal of Pxd1-YFP was visible in nearly all cells, including those with septa (Fig 2A), suggesting that Pxd1 downregulation in S phase is abolished when CRL4$^{Cdt2}$ is defective. We next performed cell cycle synchronization analysis in *ddb1Δ* and *cdt2Δ* mutants. Because the severe growth defects of *ddb1Δ* and *cdt2Δ* hindered this analysis, *spd1* deletion was used to suppress their growth defects [35,64]. Similar to the situation in wild-type cells, the protein level of Pxd1 decreased in *spd1Δ* cells synchronously released into S phase (Fig 2B and S2A Fig). In contrast, in *ddb1Δ spd1Δ* and *cdt2Δ spd1Δ* cells, Pxd1 level remained largely unchanged throughout the cell cycle (Fig 2B and 2C and S2A Fig).

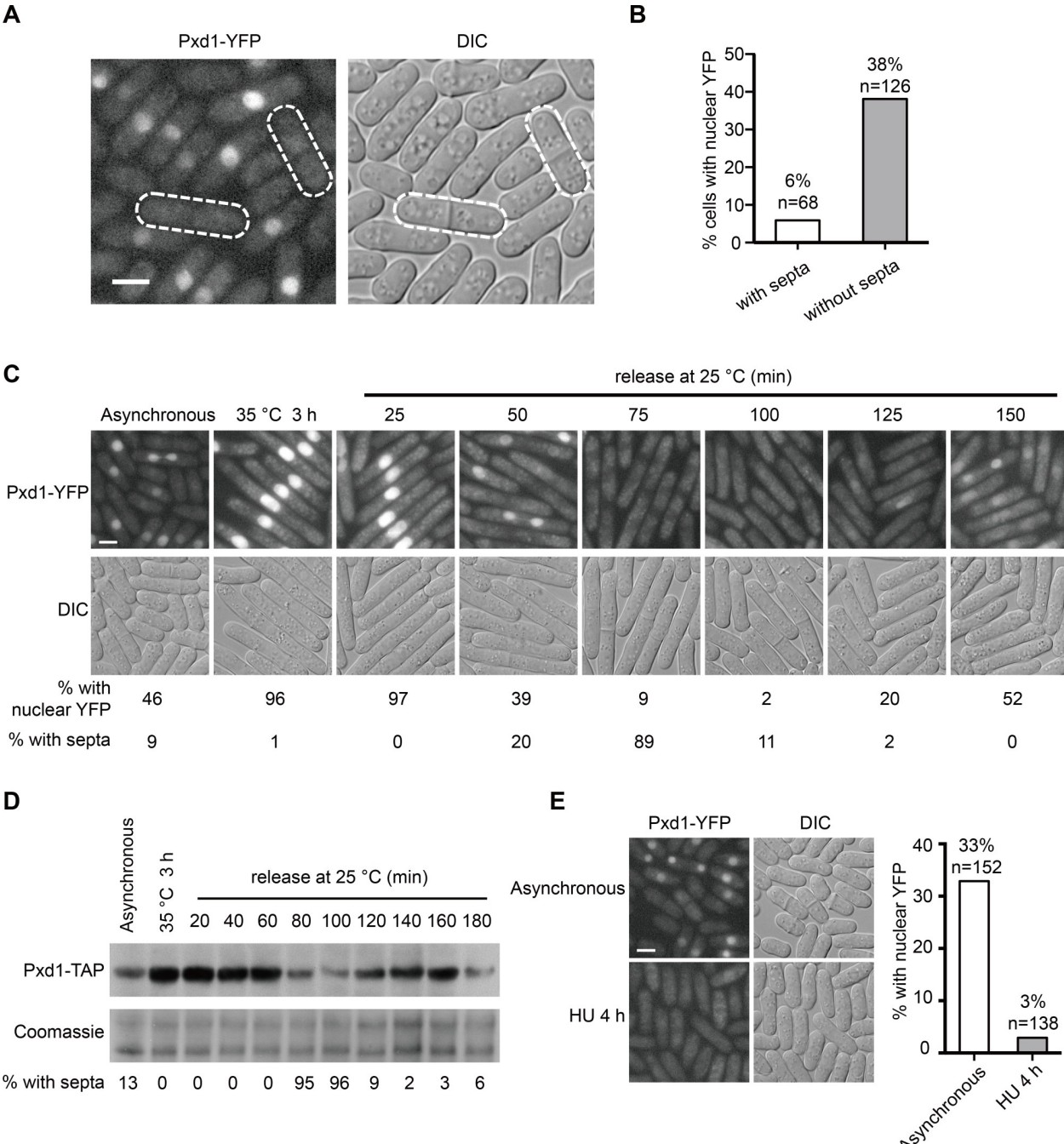

**Fig 1. The protein level of Pxd1 is reduced in S phase.** (A, B) Micrographs (A) and quantitation (B) showing that the fluorescence signal of Pxd1-YFP was more infrequently observed in septum-containing cells in an asynchronous culture. Endogenous Pxd1 was tagged with YFP. The fluorescence signal of Pxd1-YFP was observed by live cell imaging. The outlines of two cells with septa are marked by dashed lines. DIC, differential interference contrast. (C) The signal of Pxd1-YFP in cells synchronized by *cdc25-22* arrest and release. Temperature-sensitive *cdc25-22* mutant cells were arrested at G2 by incubating at 35°C for 3 h, and then released back into the cell cycle by shifting to 25°C. Cells was examined at indicated time points by live cell imaging. The percentage of cells with nuclear YFP signal and the percentage of cells with septa are shown. (D) The protein level of Pxd1-TAP in cells synchronized by *cdc25-22* arrest and release. Endogenous Pxd1 was tagged with the TAP tag and was detected by immunoblotting using the peroxidase anti-peroxidase (PAP) reagent. Coomassie staining was used as loading control. (E) Micrographs (left) and quantitation (right) showing that HU treatment reduced the percentage of cells with visible fluorescence signal of Pxd1-YFP. Cells were treated with 12 mM HU for 4 h before examined by live cell imaging. Bars, 3 μm. n, the number of cells used for quantitation.

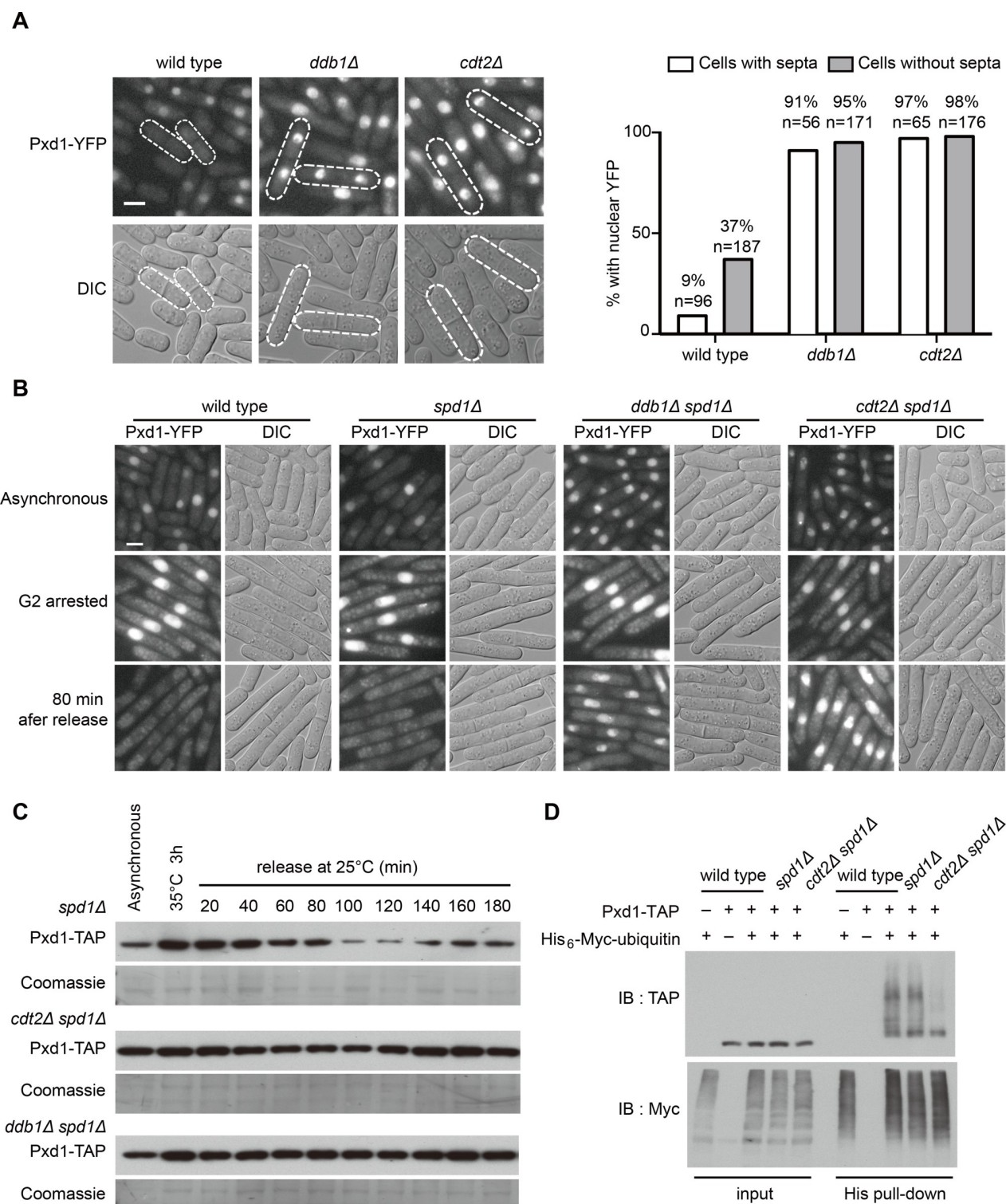

**Fig 2. CRL4^Cdt2 mediates the ubiquitination and degradation of Pxd1 in S phase.** (A) Micrographs (left) and quantitation (right) showing that the fluorescence signal of Pxd1-YFP was visible in nearly all cells in asynchronous cultures of *ddb1Δ* and *cdt2Δ* mutants. Cells with septa are marked. n, the number of cells used for the quantitation of Pxd1-YFP. (B) The fluorescence signal of Pxd1-YFP in asynchronous cells, cells arrested in G2 by *cdc25-22*, and cells released back into the cell cycle for 80 min. See S2A Fig for quantitation. (C) The protein level of Pxd1-TAP in cells synchronized by *cdc25-22* arrest and release. (D) The ubiquitination of Pxd1 is dependent on the CRL4^Cdt2 E3 ligase. Ubiquitinated proteins were enriched under denaturing condition using Ni-NTA beads. His$_6$-Myc-tagged ubiquitin and Pxd1-TAP were detected by immunoblotting (IB). Bars, 3 μm.

Moreover, we found that HU-induced reduction of Pxd1 was blocked by *ddb1Δ* and *cdt2Δ* (S2B–S2D Fig).

If CRL4$^{Cdt2}$ E3 ligase mediates the S-phase reduction of Pxd1, we reasoned that it might do so by ubiquitinating Pxd1. Indeed, using the temperature-sensitive proteasome mutant *mts2-1* to prevent the degradation of ubiquitinated proteins, we found that Pxd1 is ubiquitinated in S-phase-arrested wild-type and *spd1Δ* cells, and the ubiquitination level of Pxd1 was dramatically decreased in *cdt2Δ spd1Δ* cells (Fig 2D). These results suggest that the S-phase reduction of Pxd1 is due to CRL4$^{Cdt2}$-mediated ubiquitination and degradation of Pxd1.

## Degradation and ubiquitination of Pxd1 require PCNA and a PIP degron in Pxd1

PCNA is required for the degradation of substrates of CRL4$^{Cdt2}$ E3 ligase [39,40]. A point mutation in the *S. pombe* PCNA gene, *pcn1-D122A*, blocks the CRL4$^{Cdt2}$-mediated degradation of Cdt1 and Spd1 [40,42]. We found that the signal of Pxd1-YFP was uniformly present in *pcn1-D122A* mutant cells, including those with septa (Fig 3A and S3A Fig). Moreover, HU-induced degradation of Pxd1 was blocked by the *pcn1-D122A* mutation (Fig 3A and 3B and S3A Fig). These results indicate that the S-phase degradation of Pxd1 shares the same PCNA dependence as the degradation of previously known substrates of CRL4$^{Cdt2}$ E3 ligase.

To further investigate the mechanism of CRL4$^{Cdt2}$-mediated degradation of Pxd1, we performed a truncation analysis to identify which region of Pxd1 is important for its HU-induced degradation. This analysis showed that Pxd1(1–73) but not Pxd1(1–60) was efficiently degraded upon HU treatment (Fig 3C, S3B Fig). The degradation of Pxd1(1–73) is CRL4$^{Cdt2}$- and PCNA-dependent (S3C Fig), suggesting that amino acids 61–73 of Pxd1 contain elements needed for CRL4$^{Cdt2}$-mediated degradation. CRL4$^{Cdt2}$-mediated degradation is known to require the PIP degron, a short PCNA-interacting motif in the substrates [39]. Based on the 12-amino-acid consensus sequence of the PIP degron [39], we identified Pxd1(58–69) as a potential PIP degron (Fig 3D). It should be noted that PIP degrons previously identified in *S. pombe* Cdt2 and Spd1 all deviate substantially from the consensus sequence [41,42]. We introduced alanine substitutions into the potential PIP degron to generate two types of mutations, called PIP4A and PIP5A, respectively (Fig 3D). Both types of mutations abolished HU-induced degradation of full-length Pxd1 and Pxd1(1–73) (Fig 3E and S3D Fig), supporting that Pxd1(58–69) is a PIP degron. Mutating the last lysine residue in the PIP degron ("B+4 residue") alone impeded but did not abolish HU-induced degradation (S3D Fig), similar to the situation for Xic1 and Set8, two known substrates of CRL4$^{Cdt2}$ [40].

Consistent with the idea that the PIP degron mediates an interaction with PCNA, we found that Pxd1 interacted with Pcn1 in vivo, and this interaction was diminished by both PIP4A and PIP5A mutations (Fig 3F). It is known that CRL4$^{Cdt2}$-mediated degradation requires the substrate to bind to DNA-bound PCNA but not free PCNA [39]. The relatively mild PCNA-binding defects of Pxd1-PIP4A and Pxd1-PIP5A may reflect an ability of Pxd1 to bind to free PCNA independently of the PIP degron, an underestimation of binding to DNA-bound PCNA due to non-optimal experimental conditions, and/or a more severe defect in PCNA–PIP degron–Cdt2 complex formation. Supporting the idea that Pxd1 can bind to free PCNA and this binding only partially depends on the PIP degron, we found that Pcn1 and GST-tagged Pxd1(1–73) purified from *E. coli* interacted with each other and this interaction was diminished but not abolished by PIP4A mutations (Fig 3G). Importantly, both PIP4A and PIP5A mutations decreased the ubiquitination level of Pxd1 (Fig 3H). Together, these results strongly suggest that Pxd1(58–69) is a PIP degron and Pxd1 is a PCNA-dependent substrate of CRL4$^{Cdt2}$.

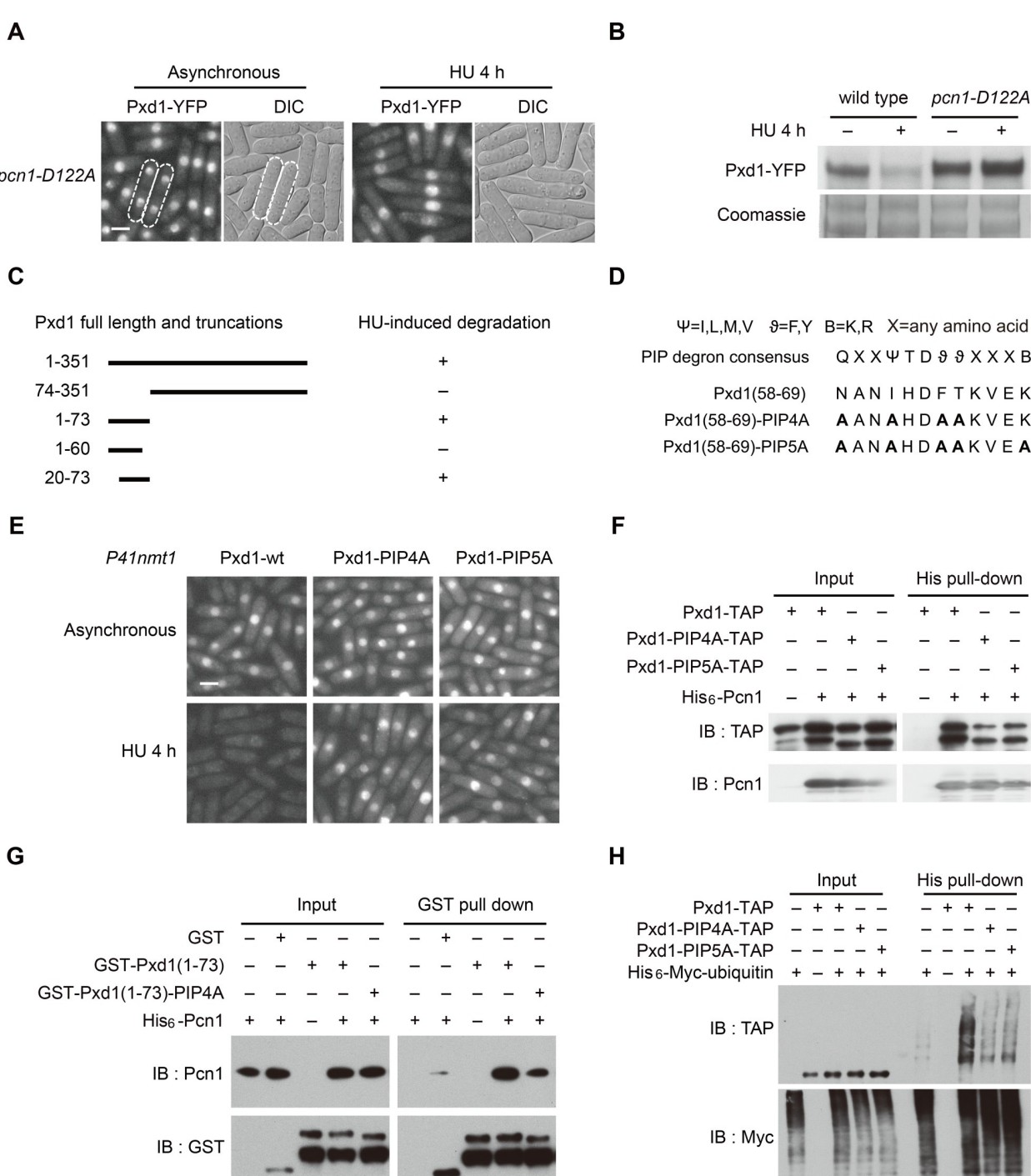

**Fig 3. Degradation and ubiquitination of Pxd1 require PCNA and a PIP degron in Pxd1.** (A) *pcn1-D122A* mutation blocked the S-phase reduction of the fluorescence signal of Pxd1-YFP. Cells with septa are marked. See S3A Fig for quantitation. (B) The protein level of Pxd1-YFP was reduced by HU treatment in wild type but not in *pcn1-D122A* mutant. (C) Summary of the results of a truncation analysis to identify which region of Pxd1 is required for its HU-induced degradation. See S3B Fig for the data. (D) The consensus sequence of the PIP degron and the sequence of the PIP degron we identified in Pxd1. PIP4A and PIP5A are two types of alanine substitution mutations we introduced into the PIP degron of Pxd1. (E) PIP4A and PIP5A mutations abolished the HU-induced reduction of the fluorescence signal of GFP-tagged Pxd1. Pxd1 was expressed from the exogenous *P41nmt1* promoter. (F) PIP4A and PIP5A mutations diminished the interaction between Pxd1 and Pcn1. TAP-tagged Pxd1 and $His_6$-tagged Pcn1 were expressed from the *Pnmt1* promoter. $His_6$-Pcn1 was pulled down using Ni-NTA beads. Pcn1 was detected by a PCNA antibody, and Pxd1 was detected by PAP. (G) PIP4A mutations diminished the in vitro interaction between recombinant Pxd1(1–73) and Pcn1. (H) PIP4A and PIP5A mutations diminished the ubiquitination of Pxd1. Enrichment of ubiquitinated proteins was performed as in Fig 2D. Bars, 3 μm.

## Non-degradable Pxd1 interferes with the S-phase functions of Dna2

Dna2 participates in Okazaki fragment maturation and the processing of stalled DNA replication forks in S phase, and is essential for cell proliferation [15,17,66]. The inhibition of Dna2 by Pxd1 [11], if not regulated, may cause problems in S phase and impede cell proliferation. We hypothesized that the degradation of Pxd1 in S phase may be a mechanism to prevent Pxd1 from hampering the S-phase functions of Dna2. Expressing the non-degradable Pxd1-PIP5A mutant in *pxd1Δ* or wild-type cells did not cause any obvious growth defect (Fig 4A and 4B), possibly because Dna2 activity in wild type cells is in excess for normal proliferation. We reasoned that the inhibition of Dna2 by non-degradable Pxd1 is more likely to be phenotypically apparent in strain backgrounds where Dna2 activity is already attenuated. Indeed, in the background of *dna2-C2*, a temperature-sensitive mutant of *dna2* [66], *pxd1-PIP5A* caused lethality at the permissive temperature for *dna2-C2* (Fig 4A), supporting the idea that a failure to degrade Pxd1 in S phase can impair Dna2 functions, and when Dna2 itself is already defective, result in cell proliferation failure.

Dna2 is known to be required for processing and restarting replication forks perturbed by the topoisomerase I poison camptothecin (CPT) [19,67,68]. Consistent with this, we found that *dna2-C2* was sensitive to CPT at its permissive temperature (S4A Fig). During our experiments, we happened to notice that a TAP-tagged allele of *dna2* (*dna2-TAP*) did not affect cell growth but caused a mild CPT sensitivity, which is weaker than that of *dna2-C2* at its permissive temperature (S4A Fig), indicating that *dna2-TAP* is a partial loss-of-function mutant with a weaker defect than *dna2-C2* at its permissive temperature. Remarkably, *pxd1-PIP5A* was lethal in the *dna2-TAP* background (Fig 4B). This result suggests that Dna2 inhibition caused by non-degradable Pxd1 is intolerable to cells with even a mildly defective Dna2.

To verify that the synthetic lethality between *pxd1-PIP5A* and *dna2-TAP* is due to inhibition of Dna2 by non-degradable Pxd1, we introduced into *pxd1-PIP5A* separation-of-function mutations that abolish only its Dna2 inhibition ability or only its Rad16 activation ability [11]. As expected, the *Δ(302–348)* mutation that abrogates Dna2 inhibition rescued the synthetic lethality, whereas the *Δ(108–226)* mutation that abrogates Rad16 activation did not (Fig 4C and S4B Fig). It has been reported that the temperature-sensitive phenotype of *dna2-C2* can be suppressed by reducing the activity of the DNA helicase Pfh1, because the need for Dna2 in Okazaki fragment maturation is reduced when long flaps generated by Pfh1 are decreased [69]. We found that *pfh1-R23*, a partial loss-of-function mutant of *pfh1*, can rescue the synthetic lethality between *pxd1-PIP5A* and *dna2-TAP* (Fig 4D), indicating that the lethal effect of non-degradable Pxd1 in the *dna2-TAP* background is due to a failure in Okazaki fragment maturation. Together, these genetic interaction data suggest that S-phase degradation of Pxd1 prevents its harmful inhibition of Dna2 that can lead to defective Okazaki fragment maturation and lethality in a Dna2 partial loss-of-function background (Fig 4E).

The fact that the *dna2-TAP* strain has no growth defect but is mildly sensitive to CPT led us to hypothesize that even though non-degradable Pxd1 does not cause an obvious growth defect, it may result in CPT sensitivity. Indeed, we found that the *pxd1-PIP5A* mutant exhibited a moderate sensitivity to CPT (Fig 4F). Introducing into *pxd1-PIP5A* the *Δ(302–348)* mutation that disrupts the Dna2 inhibition ability largely suppressed the CPT sensitivity, whereas introducing the *Δ(108–226)* mutation that disrupts the Rad16 activation ability slightly exacerbated the CPT sensitivity (Fig 4F). These data indicate that S-phase degradation of Pxd1 is important for a normal level of CPT resistance, likely because an unhindered activity of Dna2 is needed for cells to combat CPT-induced replication stress. The exact cause of the CPT sensitivity of the *pxd1-PIP5A* mutant remains to be determined and it is possible that this phenotype is due to a repair defect rather than a replication-related defect.

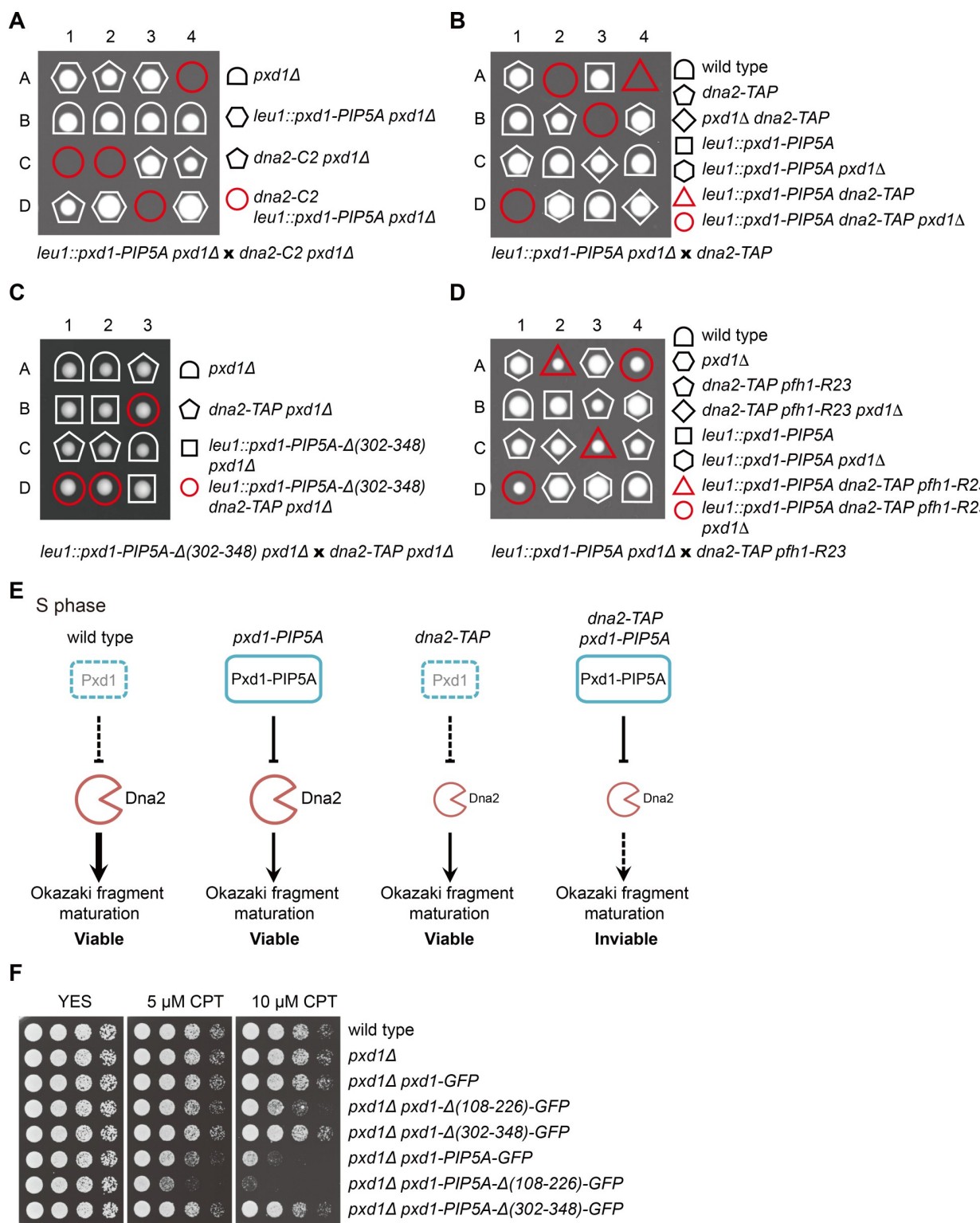

**Fig 4. Non-degradable Pxd1 interferes with the S-phase functions of Dna2.** (A) Tetrad analysis showing that *pxd1-PIP5A* is synthetic lethal with *dna2-C2*. The four spore clones of a tetrad are labeled A, B, C, and D. (B) Tetrad analysis showing that *pxd1-PIP5A* is synthetic lethal with *dna2-TAP*. (C) The synthetic lethality between *pxd1-PIP5A* and *dna2-TAP* was rescued by the Pxd1 truncation mutation that abrogates Dna2 inhibition. (D) The synthetic lethality between *pxd1-PIP5A* and *dna2-TAP* was rescued by *pfh1-R23*. *dna2-TAP* and *pfh1-R23* are genetically linked. (E) Model explaining the synthetic lethality between *pxd1-PIP5A* and *dna2-TAP*. (F) *pxd1-PIP5A* caused CPT sensitivity and this sensitivity was suppressed by the truncation of Pxd1 that abrogates Dna2 inhibition. Serial dilutions of strains were spotted on YES plates without and with CPT.

## Loss of Cdt2 interferes with the S-phase functions of Dna2

Because CRL4$^{Cdt2}$ is required for the degradation of Pxd1 in S phase, we expected that disabling CRL4$^{Cdt2}$ should compromise S-phase Dna2 functions to the same extent as the non-degradable Pxd1 and, consequently, that CRL4$^{Cdt2}$ mutants may share the same genetic interactions as the *pxd1-PIP5A* mutant. Indeed, we found that both *cdt2Δ and ddb1Δ* were synthetic lethal with *dna2-TAP* (Fig 5A and S5A Fig). Furthermore, *pxd1* deletion rescued the synthetic lethality (Fig 5A and S5A Fig), supporting the idea that the synthetic lethality is due to the lack of Pxd1 degradation. Reintroducing into *cdt2Δ dna2-TAP pxd1Δ* mutant truncated Pxd1 with only Dna2-inhibition activity resulted in lethality again, whereas reintroducing truncated Pxd1 with only Rad16-activation activity had no effect (Fig 5B and S5B Fig). These results suggest that stabilized Pxd1 in *cdt2Δ* interferes with the normal functions of Dna2, and causes lethality in a Dna2 partial loss-of-function background (S5C Fig).

As mentioned earlier, the deletion of *spd1* largely rescued the growth defect of *cdt2Δ*. However, we observed that the *cdt2Δ spd1Δ* double mutant still exhibited CPT sensitivity (Fig 5C). Because non-degradable Pxd1 causes CPT sensitivity (Fig 4F), we hypothesized that the CPT sensitivity of *cdt2Δ spd1Δ* is at least partly due to the lack of Pxd1 degradation. Consistent with this idea, the deletion of *pxd1* partially suppressed the CPT sensitivity of *cdt2Δ spd1Δ* (Fig 5C). Reintroducing truncated Pxd1 with only Dna2-inhibition activity but not truncated Pxd1 with only Rad16-activation activity abolished the suppression effect (Fig 5D). These results support the idea that CRL4$^{Cdt2}$-mediated degradation of Pxd1 is important for Dna2 to normally carry out its S-phase functions.

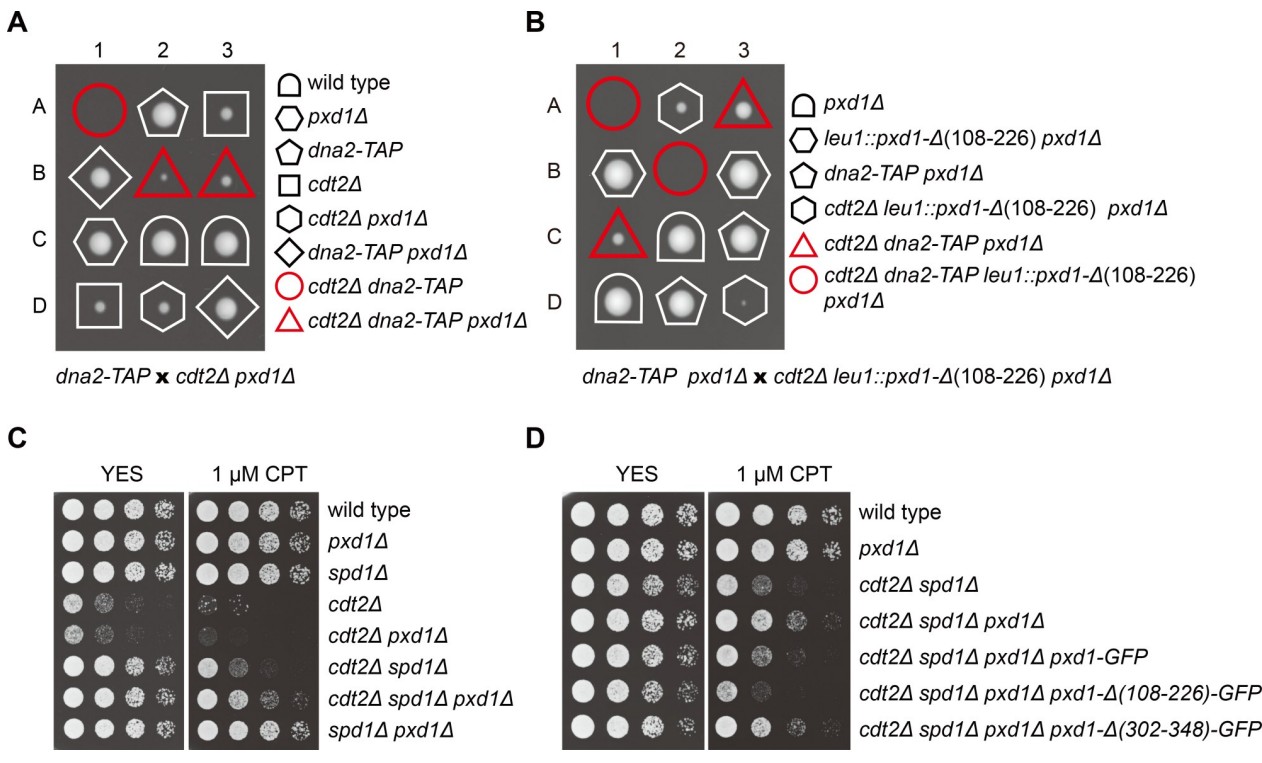

**Fig 5. Loss of Cdt2 interferes with the S-phase functions of Dna2.** (A) *cdt2Δ* was synthetic lethal with *dna2-TAP* and this synthetic lethality was suppressed by the deletion of *pxd1*. (B) Introducing truncated Pxd1 with only Dna2-inhibition activity into the *cdt2Δ dna2-TAP pxd1Δ* mutant resulted in lethality. (C) The deletion of *pxd1* suppressed the CPT sensitivity of the *cdt2Δ spd1Δ* mutant. Serial dilutions of strains were spotted on YES plates without and with CPT. (D) Introducing truncated Pxd1 with only Dna2-inhibition activity but not truncated Pxd1 with only Rad16-activation activity enhanced the CPT sensitivity of the *cdt2Δ spd1Δ pxd1Δ* mutant.

## CRL4<sup>Cdt2</sup>-dependent Pxd1 degradation decreases the nuclease activity of Rad16 in S phase

Pxd1 is an activator of the nuclease activity of Rad16 [11]. The degradation of Pxd1 in S phase is expected to result in a decrease of the nuclease activity of Rad16 in S phase. To test this idea, we measured the nuclease activity of Rad16 immunopurified from cells in which the only form of Pxd1 is Pxd1-Δ(302–348) so as to avoid co-purifying Dna2 (Fig 6A). The nuclease activity of

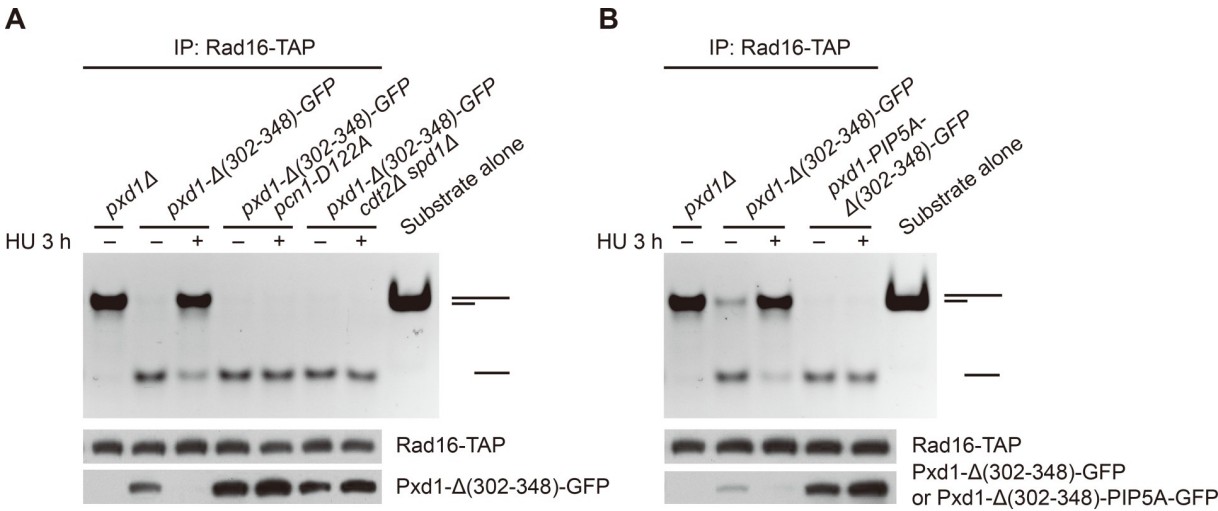

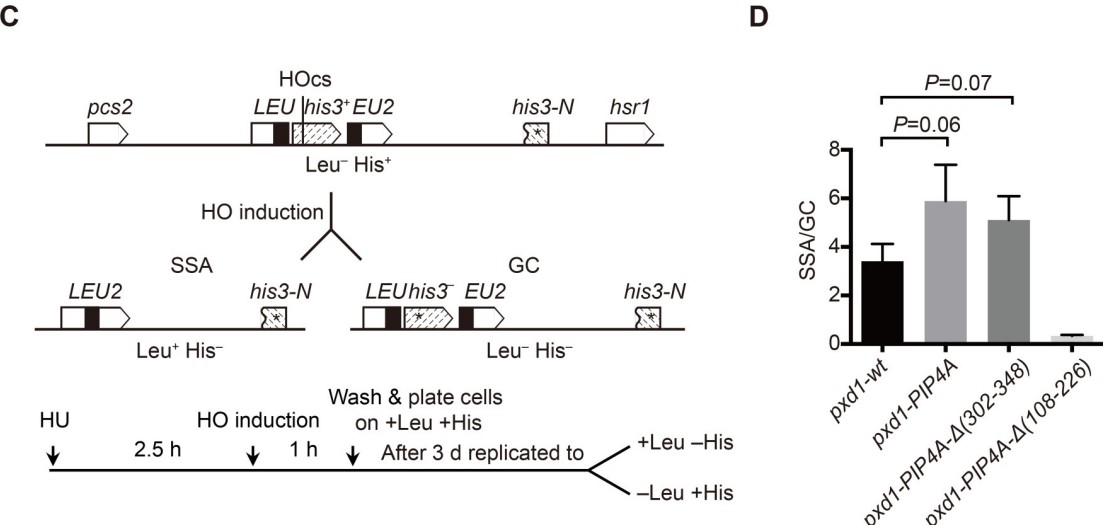

**Fig 6. CRL4<sup>Cdt2</sup>-mediated degradation of Pxd1 reduces Rad16 nuclease activity and restrains SSA in S phase.** (A) The nuclease activity of Rad16 was reduced in HU-arrested S-phase cells and this reduction was blocked in *pcn1-D122A* and *cdt2Δ spd1Δ* mutants. Rad16-TAP was immunoprecipitated with IgG beads from indicated strains expressing Pxd1-Δ(302–348), and then incubated with a 3′ overhang DNA substrate for 1 h at 30˚C. After separation by a 10% native PAGE gel, the reaction products were stained by ethidium bromide. (B) *pxd1-PIP5A* mutations abolished the reduction of the nuclease activity of Rad16 in HU-arrested S-phase cells. (C) Schematic of the SSA/GC competition assay. The SSA/GC competition system was constructed as described in Materials and Methods. HOcs, HO cleavage site. *LEU* and *EU2*, two fragments of the *S. cerevisiae LEU2* gene with an approximately 500-bp overlapping sequence. *his3-N*, a GC donor sequence including the sequence upstream of the HO site, a stop codon (denoted as an asterisk) in place of the HO cleavage site, and the N-terminal coding sequence of *his3*. In this assay, after HO induction, the HO-generated DSB can be repaired by SSA or GC. if SSA is used, cells are converted from Leu<sup>−</sup> His<sup>+</sup> to Leu<sup>+</sup> His<sup>−</sup>; if GC is used, cells are converted from Leu<sup>−</sup> His<sup>+</sup> to Leu<sup>−</sup> His<sup>−</sup>. To monitor the repair choice in S phase, cells were synchronized by HU before HO induction. (D) *pxd1-PIP4A* mutations increased the ratio between SSA and gene conversion (SSA/GC), and this increase required the Rad16 activating ability of Pxd1, but not its Dna2 inhibiting ability. Mean ± SD is shown. *P* value was calculated by unpaired Student's t test.

Rad16 purified from cells arrested in S phase by HU was indeed markedly lower than that of Rad16 purified from asynchronous cells (Fig 6A). Consistent with the idea that the lower Rad16 activity in S phase is due to PCNA- and CRL4$^{Cdt2}$-dependent degradation of Pxd1, the decrease of Rad16 activity in HU-arrested cells was not observed for either *pcn1-D122A* mutant or *cdt2Δ spd1Δ* mutant (Fig 6A). Similarly, when we introduced the PIP degron mutations into Pxd1-Δ(302–348) to block its S-phase degradation, the nuclease activity of Rad16 no longer decreased upon HU arrest (Fig 6B). These results suggest that the nuclease activity of Rad16 is temporally regulated in the cell cycle by CRL4$^{Cdt2}$-mediated degradation of Pxd1.

## CRL4$^{Cdt2}$-mediated degradation of Pxd1 restrains SSA

DSBs flanked by direct repeats can be repaired by SSA, an unfaithful and dangerous repair pathway that results in the loss of the intervening sequence between the repeats [25]. Efficient SSA in *S. pombe* requires Pxd1, as Pxd1-stimulated Rad16 nuclease activity is needed for the removal of nonhomologous single-stranded DNA tails during SSA [11,14]. We hypothesized that CRL4$^{Cdt2}$-mediated degradation of Pxd1 may be a mechanism to suppress the usage of SSA so as to bias the repair pathway choice in favor of a safer alternative. To test this idea, we constructed a repair pathway competition system modified from an SSA system developed by the Carr lab [70,71]. In this repair pathway competition system (Fig 6C), an HO-nuclease-generated DSB adjacent to the *his3* coding sequence can be repaired either by SSA using the repeat sequence in two *LEU2* fragments flanking the DSB, or by gene conversion (GC) using an inverted donor sequence 8.6 kb away. Both types of repair convert cells from His$^+$ to His$^-$, but only SSA converts cells from Leu$^-$ to Leu$^+$. The ratio of SSA vs. GC repair outcomes can be determined by dividing the number of His$^-$ Leu$^+$ colonies by the number of His$^-$ Leu$^-$ colonies. Because the effect of blocking CRL4$^{Cdt2}$-mediated Pxd1 degradation is maximal in S-phase cells, we synchronized cells in S phase before inducing HO expression. We found that the SSA/GC ratio increased 73% when CRL4$^{Cdt2}$-mediated Pxd1 degradation was abolished by *pxd1-PIP4A* mutations (*P* = 0.06, Student's t test) (Fig 6D), supporting the idea that SSA is limited by CRL4$^{Cdt2}$-mediated Pxd1 degradation in wild-type cells. As expected from the critical importance of Pxd1-mediated Rad16 activation in SSA, introducing into *pxd1-PIP4A* the *Δ (108–226)* mutation that abrogates Rad16 activation dramatically decreased the SSA/GC ratio (Fig 6D). In contrast, introducing into *pxd1-PIP4A* the *Δ(302–348)* mutation that abrogates Dna2 inhibition did not alter the SSA/GC ratio. These results indicate that CRL4$^{Cdt2}$-mediated degradation of Pxd1 restrains the usage of the SSA pathway by downregulating Rad16 nuclease activity, and thus reduces the risk of SSA-mediated chromosomal deletions.

## Discussion

To maintain genome stability and prevent unwanted DNA breaks, the activities of SSEs must be tightly coordinated with cell cycle and DNA damage response [3,4]. Previous studies showed that cell cycle kinases [72–77] and checkpoint kinases [17,74,78,79] play key roles in the regulation of SSEs. Here we demonstrate that CRL4$^{Cdt2}$ E3 ubiquitin ligase regulates two SSEs Dna2 and Rad16 (XPF) by targeting their regulator Pxd1 for degradation in S phase, thus uncovering a new mode of SSE regulation. In wild-type *S. pombe* cells, PCNA-dependent and CRL4$^{Cdt2}$-mediated degradation of Pxd1 in S phase promotes the S-phase functions of Dna2 and restrains the SSA function of Rad16, and thus helps to maintain genome stability (Fig 7). In CRL4$^{Cdt2}$-defective mutants, failure to degrade Pxd1 in S phase interferes with the S-phase functions of Dna2 and undermines the ability of cells to cope with replication stress, and in the meantime, unchecked Rad16 increases the risk of SSA-mediated genomic deletions (Fig 7). We showed previously that Pxd1 shares sequence and functional similarities with metazoan

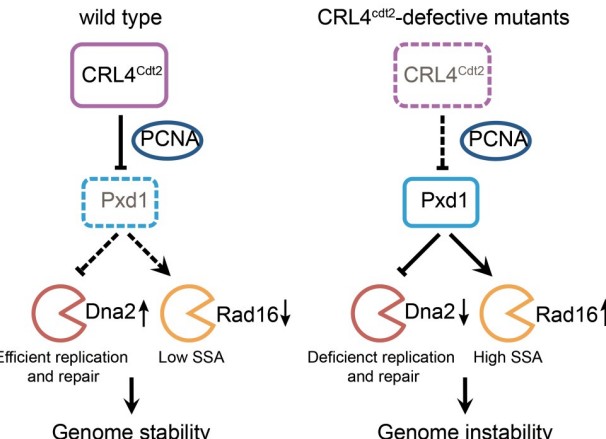

**Fig 7. Model explaining how CRL4$^{Cdt2}$ ubiquitin ligase promotes genome stability by regulating the activities of Dna2 and Rad16 through PCNA-dependent degradation of Pxd1 in S phase.**

Slx4 proteins [11]. Even though the PIP degron in Pxd1 does not reside in a region conserved in metazoan Slx4 proteins, it is still possible that some aspects of the regulatory mechanisms found here in *S. pombe* are conserved in higher eukaryotes.

As a key molecule involved in multiple DNA metabolism processes, Dna2 is known to be controlled by a myriad of regulatory mechanisms including post-translational modifications and protein-protein interactions [80]. The previously best understood mechanism of cell cycle regulation of Dna2 is in *S. cerevisiae*, where Cdk1 phosphorylation of Dna2 promotes its targeting to DSBs and enhances long-range resection in S and G2 phases [77]. Here we uncovered in *S. pombe* a mechanism specifically enhancing the S-phase functions of Dna2. Such a mechanism maximizes the activity of Dna2 at exactly the cell cycle phase where Dna2 is most intensely needed. It may not be a coincidence that another substrate of CRL4$^{Cdt2}$ in *S. pombe*, Spd1, is also an inhibitor of an enzyme whose activity is most heavily needed in S phase. There may be yet to be discovered mechanisms where S-phase upregulation of an enzymatic activity is achieved through CRL4$^{Cdt2}$-mediated degradation of an enzyme inhibitor.

When a DSB is generated at a genomic position flanked by direct repeats, homologous recombination repair of the DSB can occur either through the error-free gene conversion (GC) pathway that uses the sister chromatid as a repair template, or through the mutagenic single-strand annealing (SSA) pathway that recombines two repeats. When facing a choice between these two pathways, cells should favor GC over SSA to avoid genomic deletions. A known mechanism of SSA suppression is the inhibition of extensive DNA resection [25]. In mammalian cells, 53BP1 plays a key role in inhibiting long-range resection and restraining SSA [81]. In *S. pombe*, the Dna2- and Rqh1-dependent branch of long-range resection is inhibited by Pxd1 [11], the 53BP1 ortholog Crb2 [82], and Rad52 [83], suggesting that *S. pombe* cells employ a multi-pronged strategy to inhibit Dna2-mediated resection and consequently restrict SSA. This study uncovered a previously unknown mechanism of restraining SSA: downregulating the nuclease activity of Rad16 (XPF), which is required for the removal of nonhomologous 3′ tails during SSA. This SSA-restricting mechanism may be especially important when the intervening sequence between the repeats is short or when DNA resecting activities are abnormally high.

In S phase, CRL4$^{Cdt2}$-mediated degradation of Pxd1 disables one of the mechanisms inhibiting Dna2-mediated resection and exposes cells to an increased risk of SSA. However, the same degradation event also results in the downregulation of Rad16 activity and imposes a constraint at another step of SSA. Thus, the opposite regulation of two SSA-promoting factors, Dna2 and Rad16, by Pxd1 allows fission yeast cells to maintain a tight restriction on SSA when upregulating the activity of Dna2 during DNA replication. It will be interesting to explore whether similar strategies are used in other organisms to maintain stringent SSA suppression throughout the cell cycle.

## Materials and methods

### Fission yeast strains

The fission yeast strains used in this study are listed in S1 Table, and plasmids used in this study are listed in S2 Table. Genetic methods for strain construction and the composition of media are as described [84]. The gene conversion (GC) and single strand annealing (SSA) competition assay system was modified from the inducible SSA system created by Tony Carr's lab [70,71]. In this inducible SSA system, an HO cleavage site (HOcs) is inserted into the *his3* coding sequence and flanked by two fragments of the *S. cerevisiae LEU2* gene, called *LEU* and *EU2*. *LEU* and *EU2* share an approximately 500-bp overlapping sequence, which serves as the donor repeat for SSA repair. To construct the SSA/GC competition system, a GC donor sequence (referred to as *his3-N*) including the sequence upstream of the HO site, a stop codon in place of the HO cleavage site, and the N-terminal coding sequence of *his3* was inserted in the reverse direction at a location 8.6 kb away from the HO cleavage site. The coding sequence of the HO endonuclease was placed under the control of the uracil-inducible *Purg1* promoter using Cre-mediated cassette exchange [71].

### Cell cycle synchronization

Strains carrying the *cdc25-22* allele were cultured at 25˚C, synchronized at G2 phase by incubating at 35˚C for 3 h, and released back into the cell cycle by shifting to 25˚C.

### Fluorescence microscopy

Log-phase cells grown in EMM medium were used for examining the localization of fluorescent protein-tagged Pxd1. Microscopy was performed on a DeltaVision PersonalDV system (Applied Precision) equipped with an mCherry/YFP/CFP filter set (Chroma 89006 set) and a Photometrics CoolSNAP HQ2 camera. Images were processed using the SoftWoRx software.

### Pull down of His-tagged Pcn1

The lysate from 50 OD$_{600}$ units of cells was prepared by glass beads beating in lysis buffer (50 mM sodium phosphate, pH 8.0, 0.1 M NaCl, 10% glycerol, 0.05% Tween-20, 10 mM imidazole, 1 mM PMSF, 1 mM DTT, 1× Roche protease inhibitor cocktail). After incubating the lysate with Ni-NTA beads for 3 h and washing the beads for three times with wash buffer (50 mM sodium phosphate, pH 8.0, 0.15 M NaCl, 10% glycerol, 0.05% Tween-20, 20 mM imidazole, 1 mM DTT), proteins were eluted from the beads by boiling in 1x SDS loading buffer.

### GST pull down assay

His-tagged Pcn1 and His-GST-tagged Pxd1 were expressed in *E. coli* BL21 strain, and purified using Ni-NTA beads. Purified recombinant proteins were mixed together in binding buffer (50 mM Tris-HCl, pH 8.0, 0.1 M NaCl, 10% glycerol, 0.05% NP-40, 1 mM PMSF, 1 mM DTT),

and then incubated with Glutathione Sepharose beads for 3 h at 4˚C. After washing for three times, proteins were eluted from the beads by boiling in 1x SDS loading buffer.

## Monitoring the ubiquitination of Pxd1

Ubiquitin tagged with His$_6$-Myc was expressed from the *Pnmt1* promoter. TAP-tagged Pxd1 was expressed from the *Pnmt81* promoter. Cells with the *nda3-KM311 mts2-1* genetic background were synchronized in M phase by incubation at 20˚C for 4 h, then transferred to 37˚C and incubated for 2 h with 12 mM HU. About 30 OD600 units of cells were treated with 500 μl of 1.85 N NaOH and 7.4% β-mercaptoethanol on ice for 10 min, and then incubated for anther 10 min on ice after the addition of 500 μl 50% trichloroacetic acid. The pellet was collected by centrifugation and washed for three times with ice-cold acetone, and then resuspended in 1 ml denaturing buffer (6 M guanidine-HCl, 50 mM sodium phosphate, 10 mM Tris-HCl, pH 8.0, 0.05% Tween-20, 15 mM imidazole). The suspension was incubated for 1 h on rotator at room temperature, and the pH was adjusted to ~8.0 by the addition of 1 M Tris base. After centrifugation, the supernatant was transferred to a fresh tube with 20 μl Ni-NTA beads. After overnight incubation, the beads were washed for three times with 1 ml wash buffer (8 M urea, 100 mM sodium phosphate, 10 mM Tris-HCl, pH 8.0, 0.05% Tween-20, 15 mM imidazole). Proteins were eluted from the beads by boiling the beads with 50 μl HU buffer (8 M urea, 200 mM Tris-HCl, pH 6.8, 1 mM EDTA, 5% SDS, 0.1% bromophenol blue, 1.5% dithiothreitol) at 95˚C for 10 min.

## Rad16 nuclease assay

50 OD600 units of cells treated with or without 12 mM HU were collected and lysed with 400 μl lysis buffer (50 mM Tris-HCl, pH 8.0, 0.1 M NaCl, 10% glycerol, 0.05% NP-40, 1 mM PMSF, 1 mM DTT, 1× Roche protease inhibitor cocktail). Rad16-TAP immunoprecipitated by IgG beads was incubated with the 3′ overhang substrate in 20 μl reaction buffer (50 mM Tris-HCl, pH 7.5, 50 mM NaCl, 1 mM MnCl$_2$, 1 mM dithiothreitol, 0.1 mg/ml BSA) at 30˚C for 1 h. Reaction products were separated on 10% native PAGE gels. The native PAGE gels were stained with EB. Oligo461 and oligo462 were used to generate the 3′ overhang substrate as previously described [11].

## Single strand annealing (SSA) and gene conversion (GC) competition assay

To examine how the S-phase degradation of Pxd1 may affect repair pathway choice, we arrested the cells in S phase by HU treatment for 2.5 h. After the addition of 1x uracil and histidine, the cells were incubated for another 1 h at 30˚C for inducing the expression of HO endonuclease, and then plated on plates containing leucine and histidine, and incubated at 30˚C for 3 days. Colonies formed on plates were replicated to −His plates and −Leu plates, respectively. Cells that had repaired the HO-induced double-strand breaks (DSBs) by either SSA or GC became His⁻ and could not grow on −His plates. Cells that had repaired the DSBs by SSA became Leu⁺ and could grow on −Leu plates. Cells that had repaired the DSBs by GC remained to be Leu⁻ and could not grow on −Leu plates. The ratio of SSA vs. GC repair outcomes was calculated by dividing the number of His⁻ Leu⁺ colonies by the number of His⁻ Leu⁻ colonies.

## Supporting information

**S1 Fig. The protein level of Pxd1 is reduced in S phase.** (A) The protein level of Pxd1-YFP expressed from the *P81nmt1* promoter or its own promoter was examined in asynchronous cells and HU-treated cells. Cells were treated with 12 mM HU for 2 h or 4 h. (B) Micrographs

(left) and quantitation (right) showing that HU treatment reduced the fluorescence signal of Pxd1-YFP expressed from the *P81nmt1* promoter. Bar, 3 μm. n, the number of cells used for quantitation.
(TIF)

**S2 Fig. CRL4<sup>Cdt2</sup> mediates the ubiquitination and degradation of Pxd1 in S phase.** (A) Quantitation of the live cell imaging data shown in Fig 2B. The percentage of cells with nuclear YFP signal and the percentage of cells with septa are shown. n, the number of cells used for quantitation. (B-C) Micrographs (B) and quantitation (C) showing that HU treatment did not affect the fluorescence signal of Pxd1-YFP in *ddb1Δ* and *cdt2Δ* cells. Bar, 3 μm. (D) The protein level of Pxd1-TAP in asynchronous cells and HU-treated cells of wild type and the *cdt2Δ* mutant.
(TIF)

**S3 Fig. Degradation and ubiquitination of Pxd1 requires PCNA and a PIP degron in Pxd1.** (A) Quantitation of the live cell imaging data shown in Fig 3A. The percentage of cells with nuclear YFP signal is shown. n, the number of cells used for quantitation. (B) The fluorescence signals of GFP-tagged full-length Pxd1 (351 amino acids) and truncated Pxd1 fragments. The expression was driven by the *P41nmt1* promoter. The results are summarized in Fig 3C. (C) The fluorescence signal of Pxd1(1–73)-GFP in asynchronous and HU-treated cells of wild type, *pxd1Δ*, *cdt2Δ spd1Δ*, and *pcn1-D122A* mutants. The expression was driven by the *P41nmt1* promoter. (D) The fluorescence signal of GFP-tagged Pxd1(1–73), Pxd1(1–73)-PIP4A, Pxd1(1–73)-K69A, and Pxd1-PIP5A in asynchronous and HU-treated cells. The expression was driven by the *P41nmt1* promoter. Bars, 3 μm.
(TIF)

**S4 Fig. Non-degradable Pxd1 interferes with the S-phase functions of Dna2.** (A) *dna2-C2* and *dna2-TAP* mutants exhibited sensitivity to CPT. Serial dilutions of strains were spotted on YES plates without and with CPT. (B) The synthetic lethality between *pxd1-PIP5A* and *dna2--TAP* was not rescued by the Pxd1 truncation mutation that abrogates Rad16 activation.
(TIF)

**S5 Fig. Loss of Cdt2 interferes with the S-phase functions of Dna2.** (A) *ddb1Δ* was synthetic lethal with *dna2-TAP* and this synthetic lethality was suppressed by the deletion of *pxd1*. (B) Introducing truncated Pxd1 with only Rad16-activation activity did not affect the growth of the *cdt2Δ dna2-TAP pxd1Δ* mutant. (C) Model explaining the synthetic lethality between *cdt2Δ* and *dna2-TAP*.
(TIF)

**S1 Table. Fission yeast strains used in this study.**
(PDF)

**S2 Table. Plasmids used in this study.**
(PDF)

## Acknowledgments

We are grateful to Drs. Stephen E. Kearsey, Masaru Ueno, and Tony Carr for strains and plasmids, and to members of the Du lab for helpful discussions.

## Author Contributions

**Conceptualization:** Jia-Min Zhang, Li-Lin Du.

**Funding acquisition:** Meng-Qiu Dong, Li-Lin Du.

**Investigation:** Jia-Min Zhang, Jin-Xin Zheng, Yue-He Ding, Xiao-Ran Zhang, Fang Suo, Jing-Yi Ren, Meng-Qiu Dong, Li-Lin Du.

**Writing – original draft:** Jia-Min Zhang, Li-Lin Du.

**Writing – review & editing:** Jia-Min Zhang, Li-Lin Du.

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
