## [Decision Letter · Decision Letter 0]

13 May 2020

Dear Dr Du,

Thank you very much for submitting your Research Article entitled 'CRL4Cdt2 ubiquitin ligase regulates Dna2 and Rad16 (XPF) nucleases by targeting Pxd1 for degradation' to PLOS Genetics. Your manuscript was fully evaluated at the editorial level and by three independent peer reviewers. The reviewers appreciated the attention to an important topic but identified some aspects of the manuscript that should be improved.

We therefore ask you to modify the manuscript according to the review recommendations before we can consider your manuscript for acceptance. Specifically, we ask you to address the issue raised by Reviewer #3 about the S phase function affected by stabilized Pxl1. Your revisions should address the specific points made by each reviewer.

Yours sincerely,

Olaf Nielsen

Guest Editor

PLOS Genetics

Gregory P. Copenhaver

Editor-in-Chief

PLOS Genetics

Reviewer's Responses to Questions

**Comments to the Authors:**

Reviewer #1: The authors previously identified Pxd1 as an activator of Rad16 and an inhibitor of Dna2 nuclease activities. The current manuscript is a well executed study which is also well written. Results presented expand on the previous study and suggest that CRL4Cdt2 targets Pxd1 for degradation during S-phase, promoting Dna2 function and restricting Rad16 activity.

Suggestions for improvement:

1. Could the authors comment/speculate on the mechanism of Pxd1-dependent Dna2/Rad16 regulation? Have they looked at Dna2 and Rad16 levels throughout the cell cycle in their different genetic backgrounds to see if they are affected?

2. It is difficult to analyse/review the results of the tetrad analyses. While the interpretation of the genotypes is shown in the figures, without knowing which phenotypes where scored for each colony and how, this can not be verified. Would it be possible to provide this info in the supplementary data?

3. Fig. 3F: can relatively small reduction in PCNA/Pxd1 interaction in PIP4A and Pip5A account for lack of degradation?

4. Fig. 3D: what do X and B symbols mean? There seem to be several AAs that do not conform to the consensus sequence.

5. Fig. 6 a&b: Can the authors be sure that the observed nuclease activity is from Rad16 and not from a contaminating nuclease activity? Have the authors used a control (e.g. using untagged Rad16)?

6. Could the authors discuss if they think the mechanism has been conserved (e.g. in human cells). Are there any Pxd1 homologues in (higher) eukaryotes?

Reviewer #2: In a landmark 2014 paper published in PLoS Biology (reference 11), these authors reported the discovery of Pxd1, which they showed is a scaffolding protein that activates Rad16(XPF) and inhibits Dna2 structure specific endonucleases in fission yeast. Since then a key unanswered question has been why does Pxd1 exist? There is no apparent strategic purpose in constitutively activating Rad16 or inhibiting Dna2. The current study answers this question by showing that Pxd1 in specifically degraded in S-phase, thereby allowing cell cycle regulation of Rad16 and Dna2 activities. A thorough set of genetic and biochemical studies show that degradation of Pxd1 is mediated by CRL4/Cdt2 ubiquitin ligase, which was previously shown to regulate other important activities during S-phase. As predicted, this regulation requires PCNA binding to a PCNA-binding degron motif on Pxd1. In an important set of assays, the authors show that abrogation of Pxd1 regulation by CRL4/Cdt2 ubiquitin ligase increases genome instability, most notably increasing SSA repair of a DSB.

This is an excellent paper that provides key insights into the regulation of SSEs, which is a highly topical question in the fields of DNA repair and genome protection. The technical qualities of the experiments are excellent, with appropriate controls and statistical analyses (although see note below). There is ample evidence for all the key conclusions in the manuscript. The Introduction and Discussion sections are written to a high standard, with appropriate citations and interpretations of the data.

Minor issues to be addressed:

1. The Introduction should include a mention that Rad16 and XPF are used as shorthand for the Rad16-Swi10 and XPF-ERCC1 nuclease complexes, respectively.

2. It is unclear why p-values were not included in graphs such as those in Figures 1 and 2. They are hardly necessary but should be added if easily calculated.

Reviewer #3: This paper adds Pxl1 to the list of proteins that are related to S phase function and are degraded in S phase by ubiquitylation by CRL4Cdt2. Pxl1 was previously shown to inhibit Dna2 and stimulate Rad16. The data here show clearly that inactivation of the ubiquitin ligase stabilizes Pxl1 in S phase, and a reduction in Pxl1 ubiquitylation is shown. The PIP-degron in Pxl1 required for its ubiquitylation is tracked down, and mutation of the degron stabilizes the protein and prevents its interaction with PCNA. Proteolysis of Pxl1 results in a reduction of Rad16 nuclease activity, and strains with stabilized Pxl1 show an increase in single strand annealing in an assay that measures the SSA/gene conversion ratio in the repair of a DSB. However, stabilization of Pxl1 alone isn’t shown to have a replication phenotype in a wt background, and to demonstrate an effect on viability it was necessary to combine stabilized Pxl1 with defective alleles of Dna2, which makes the biological significance of the degradation less clear.

The paper is clearly presented and the data are convincing. One limitation is that it is assumed that Okazaki fragment processing is the key S phase function that is affected by stabilized Pxl1 when combined with defective Dna2. However, this is just inferred from genetic assays looking at synthetic lethality and S phase execution is not directly examined as such. Although increased sensitivity to camptothecin is shown, it isn’t clear whether this is due to a repair defect or an effect on S phase. It would be of value to demonstrate a replication phenotype with wt Dna2 using more sensitive assays, for instance by analysing S phase progression, fork rate or fork-stalling assays. Given that Dna2 is involved in several processes as well as Okazaki fragment processing, such as stalled-fork processing and replication of ‘difficult-to-replicate’ regions, it would be interesting to know which of these is affected by Pxl1 stabilization in an otherwise wt background.

Minor point:

Some of the Pxl1-YFP images appear to show foci of Pxl1 (eg Fig 2A, cdt1D, 2B, cdt2D spd1D) – is this real and does this reflect accumulation of Pxl1 at sites of DNA damage?

**Have all data underlying the figures and results presented in the manuscript been provided?**

Reviewer #1: Yes

Reviewer #2: Yes

Reviewer #3: Yes

PLOS authors have the option to publish the peer review history of their article (what does this mean?). If published, this will include your full peer review and any attached files.

Reviewer #1: No

Reviewer #2: No

Reviewer #3: No

---

## [Editor Report · Decision Letter 1]

15 Jun 2020

Dear Dr Du,

We are pleased to inform you that your manuscript entitled "CRL4Cdt2 ubiquitin ligase regulates Dna2 and Rad16 (XPF) nucleases by targeting Pxd1 for degradation" has been editorially accepted for publication in PLOS Genetics. Congratulations!

Yours sincerely,

Olaf Nielsen

Guest Editor

PLOS Genetics

Gregory P. Copenhaver

Editor-in-Chief

PLOS Genetics

Comments from the reviewers (if applicable):

**Data Deposition**

http://datadryad.org/submit?journalID=pgenetics&manu=PGENETICS-D-20-00539R1

**Press Queries**

---

## [Editor Report · Acceptance letter]

14 Jul 2020

PGENETICS-D-20-00539R1 

CRL4Cdt2 ubiquitin ligase regulates Dna2 and Rad16 (XPF) nucleases by targeting Pxd1 for degradation 

Dear Dr Du, 

We are pleased to inform you that your manuscript entitled "CRL4Cdt2 ubiquitin ligase regulates Dna2 and Rad16 (XPF) nucleases by targeting Pxd1 for degradation" has been formally accepted for publication in PLOS Genetics! Your manuscript is now with our production department and you will be notified of the publication date in due course.

With kind regards,

Kaitlin Butler

PLOS Genetics

On behalf of:
